# [Re] Spatial-Adaptive Network for Single Image Denoising

**Sami Menteş**[1]    **Furkan Kınlı**[2]    **Barış Özcan**[3]    **Furkan Kıraç**[4]

Video, Vision and Graphics Lab

Özyeğin University

{sami.mentes[1], baris.ozcan.10097[3]}@ozu.edu.tr, {furkan.kinli[2], furkan.kirac[4]}@ozyegin.edu.tr

## Reproducibility Summary

*In this study, we present our results and experience during replicating the paper titled "Spatial-Adaptive Network for Single Image Denoising". This paper proposes novel spatial-adaptive denoising architecture for efficient noise removal by leveraging the deformable convolutions to adapt spatial information (i.e. edges and textures). We have implemented the model from scratch in PyTorch framework, and then have conducted real and synthetic noise experiments on the corresponding datasets. We have achieved to reproduce the results qualitatively and quantitatively.*

### Scope of Reproducibility

The original paper proposes an encoder-decoder structure exploiting a residual spatial-adaptive block and a context block to capture multi-scale information for achieving the state-of-the-art on real and synthetic noise removal.

### Methodology

We have implemented the model, namely *SADNet*, from scratch in PyTorch as described in the paper, and also adopted the training loop and proposed blocks from the author's code. Since the weight initialization of proposed blocks was not implicitly defined in the paper, we have decided to use the default initialization method for convolutional layers in PyTorch (*i.e.* Kaiming). Experiments have been completed on a single RTX 2080 Ti in 3 days for each, and it requires ~3GB GPU memory for training, and ~8GB CPU memory for loading the data, due to the file structure of datasets.

### Results

We have achieved to reproduce the results qualitatively and quantitatively on synthetic and noise removal tasks. SADNet has the capacity to learn to remove the synthetic and real noise in images, and it produces visually-plausible outputs even after a few epochs. Moreover, we have employed SSIM and PSNR metrics to measure the quantitative performance for all settings. The quantitative results on both tasks are on-par when compared to the reported results in the paper.

### What was easy

The code was open-source, and implemented in PyTorch, hence adopting the training loop and proposed blocks to our implementation facilitated our reproduction study. The loss function is straightforward and the architecture has a *U-Net-like* structure, so that we could achieve to implement the architecture in a fair time.

### What was difficult

Due to the lack of compatibility with the current versions of PyTorch and TorchVision and the dependency on an external CUDA implementation of deformable convolutions, we have encountered several issues during our implementation. Then, we have considered to re-implement residual spatial-adaptive block and context block from scratch for deferring these dependencies, however, we could not achieve it just by referring to the paper in limited time. Therefore, we have decided to directly use the provided blocks as in the author's code.

### Communication with original authors

We did not make any contact with the authors since we achieved to solve the issues encountered during the implementation of SADNet by examining the author's code.

Preprint. Under review.

# 1 Introduction

Recent works [2, 3, 4, 5] have shown that the previous assumption of an identically-distributed additive white Gaussian noise (AWGN) is not an accurate representation of the real noise occurring in images. Traditional denoiser architectures lack the ability to adapt textures and edges, and thus miss the details while denoising, due to the over-smoothing behaviour of CNNs. A workaround to this problem is implementing a deeper network, however, such a practice introduces a more complex model with its computational burden.

In the original paper [1], an encoder-decoder architecture consisting a residual spatial-adaptive block, namely RSAB, is proposed for removing spatially-variant and channel-dependent noise while processing larger regions in each step by utilizing deformable convolutions. As the main claim of the paper, this method produces better performance than the compared methods in given benchmark, and also for the synthetic noise removal task.

In this reproducibility report, we studied SADNet architecture for both real and synthetic noise removal in detail, which contains implementing the architecture described in the paper, running the experiments, reporting the important details about certain issues encountered during reproducing, and comparing the obtained results with the ones reported in the original paper.

# 2 Scope of reproducibility

The main idea of the paper is to present a spatial-adaptive architecture with encoder-decoder structure which captures the relevant features from the complex image content while removing real noise appearing in images. Residual spatial-adaptive block (RSAB) makes it possible to achieve this in an efficient manner.

The proposed model, namely SADNet, claims to outperform the state-of-the-art performances in SSIM and PSNR metrics with a moderate run-time. To validate these claims, we try to investigate the following questions:

- Is the implementation details described in the paper and provided code sufficient for replicating the quantitative results reported in the paper?
- Are the qualitative results visually-plausible?
- Are the replicated quantitative results competitive enough?
- Could our replication obtain a proximate denoising duration compared to the reported results in the original paper?

# 3 Methodology

We have implemented the model, namely SADNet, from scratch in PyTorch [6], as described in the paper by adopting RSAB, Context block and Offset block from the author's code. The implementation of residual blocks (ResBlock) in the author's code differs from the common residual block implementation [17] by not using the output activation. In contrast to the common practice of applying a nonlinear activation function to the output, their ResBlock implementation directly forwards its output to the next level layers. At this point, the authors handle those activations at the model scope. We also removed Batch Normalization [7] from the residual blocks as proposed by the original paper. To enhance the readability of the model structure in our implementation, we imported those activation functions back in to ResBlock.

The deformable convolutions in RSAB are implemented in CUDA, hence we used NVIDIA GPUs with the relevant CUDA driver.

For validating the reported results on real noisy images, we have implemented the data loaders, which are missing in the author's code. Furthermore, we integrated WandB [8] library to the training loop in order to track our experiments during training.

## 3.1 SADNet

SADNet is an encoder-decoder architecture with skip connections which favors spatial adaptability and large receptive field over deeper networks for the well-studied denoising task. The proposed model aims to achieve the state-of-the-art denoising performance while maintaining the computational complexity by exploiting residual spatial-adaptive block (RSAB), Context Block and Offset Block. The visual representation of SADNet architecture is shown in Figure 1, and also the structural details about our implementation of SADNet can be seen in Table 1.

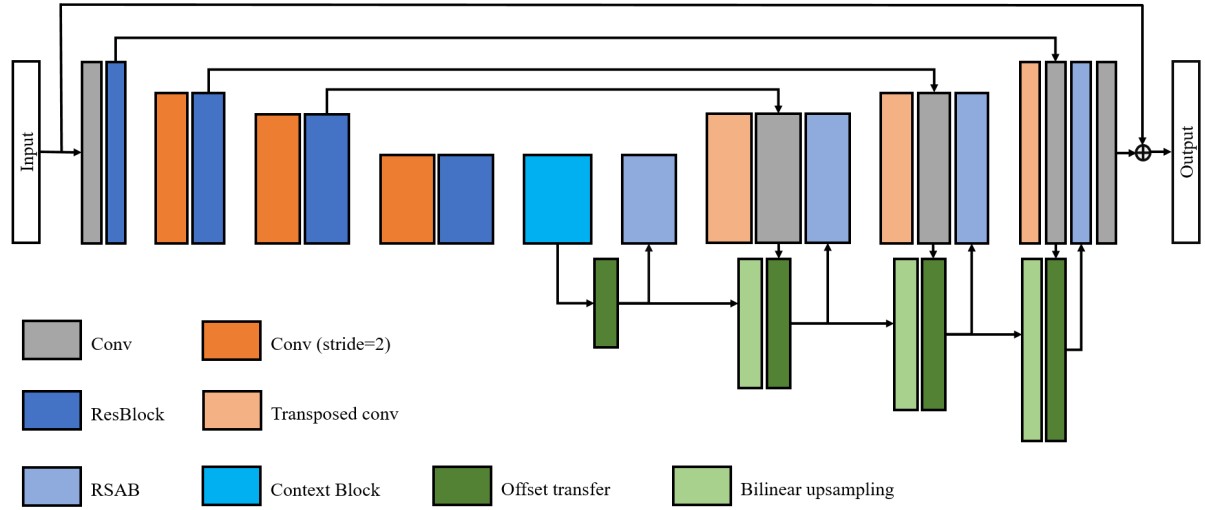

Figure 1: Representation of the SADNet architecture. Obtained from the paper [1].

Regions with the sharp texture changes in an image, typically edges and corners, raise difficulties for training the regular convolutions, due to its fixed-size weighting mechanism. Such regions, where different textures co-occur in a particular receptive field of the regular convolutions, are simply ignored during the weighting process, due to the fixed size kernels. To address this issue, self-similarity weighting is attained via modulated deformable convolutions [25] in RSAB. The kernels of the deformable convolutions have a learnable offset for each location in an image, and thus it has the capacity to adapt to the spatial texture changes. The formula of modulated deformable convolutions can be seen as follows

$$y(p) = \sum_{p_i \in N(p)} w_i \cdot x(p_i + \Delta p_i) \cdot \Delta m_i \tag{1}$$

where $\Delta p_i$ denotes the learnable offset for location $p_i$, and $\Delta m_i$ is the extra degree of freedom for adjusting the modulation scalar between $[0, 1]$.

The nature of the decoder architectures enforces to transform the feature maps from coarse to fine at each scale. For learning the offsets more accurately in RSABs, the offsets $\Delta p^{s-1}$ and the modulation scalars $\Delta m^{s-1}$ from the previous scale are further transferred into the current scale $s$ with the help of Offset Blocks. The offset transfer is formulated as

$$(\Delta p^s, \Delta m^s) = F_{offset}\left(x, F_{up}\left((\Delta p^{s-1}, \Delta m^{s-1})\right)\right) \tag{2}$$

where $F_{up}$ denotes the up-sampling operation. RSAB receive the extracted features and the reconstructed features from the previous scale conveyed by the Offset Block. The inputs are then fused through a modulated deformable convolution layer with a subsequent regular convolution layer. Moreover, a skip connection similar to ResBlock is employed to enhance the information transferring. At this point, RSAB can be formulated as,

$$F_{RSAB}(x) = F_{cn}(F_{act}(F_{dcn}(x))) + x \tag{3}$$

where $F_{cn}$ and $F_{dcn}$ denote regular convolution and modular deformable convolution, respectively. Lastly, $F_{act}$ stands for the leaky ReLU activation function [15] with a negative slope of $0.2$.

Another introduced block is the Context Block, which resides at the bottleneck of the model. To increase the size of the receptive fields while preserving the spatial resolution, Context Block is employed for the model, just between the encoder and decoder structures. Furthermore, unlike the common implementation of Context Block, Batch Normalization layer is removed in the published code, and only four dilation rates are used, which are 1, 2, 3 and 4.

Following the original paper, we used $L_1$ loss for training our model on real-noise image datasets, and $L_2$ loss for training on synthetic image datasets.

Table 1: Details of our model implementation.

| Module Name | Kernel Size | # of Channels | Stride | Non-linearity |
|---|---|---|---|---|
| Conv1 | $1 \times 1$ | $3 \to 32$ | 1 | Leaky ReLU(0.2) |
| ResBlock1 | $3 \times 3$ | $32 \to 32$ | 1 | Leaky ReLU(0.2) |
| Conv2 | $2 \times 2$ | $32 \to 64$ | 2 | Leaky ReLU(0.2) |
| ResBlock2 | $3 \times 3$ | $64 \to 64$ | 1 | Leaky ReLU(0.2) |
| Conv3 | $2 \times 2$ | $64 \to 128$ | 2 | Leaky ReLU(0.2) |
| ResBlock3 | $3 \times 3$ | $128 \to 128$ | 1 | Leaky ReLU(0.2) |
| Conv4 | $2 \times 2$ | $128 \to 256$ | 2 | Leaky ReLU(0.2) |
| ResBlock4 | $3 \times 3$ | $256 \to 256$ | 1 | Leaky ReLU(0.2) |
| Context Block | | $256 \to 256$ | | Leaky ReLU(0.2) |
| Offset Block Bottleneck | | $256 \to 256$ | | Leaky ReLU(0.2) |
| RSAB Bottleneck | | $256 \to 256$ | | Leaky ReLU(0.2) |
| TransposeConv1 | $2 \times 2$ | $256 \to 128$ | 2 | Leaky ReLU(0.2) |
| Concat1 (/w ResBlock3) | | $128 \to 256$ | | |
| Conv5 | $1 \times 1$ | $256 \to 128$ | 1 | Leaky ReLU(0.2) |
| Offset Block 1 | | $128 \to 128$ | | Leaky ReLU(0.2) |
| RSAB 1 | | $128 \to 128$ | | Leaky ReLU(0.2) |
| TransposeConv2 | $2 \times 2$ | $128 \to 64$ | 2 | Leaky ReLU(0.2) |
| Concat2 (/w ResBlock2) | | $64 \to 128$ | | |
| Conv6 | $1 \times 1$ | $128 \to 64$ | 1 | Leaky ReLU(0.2) |
| Offset Block 2 | | $64 \to 64$ | | Leaky ReLU(0.2) |
| RSAB 2 | | $64 \to 64$ | | Leaky ReLU(0.2) |
| TransposeConv3 | $2 \times 2$ | $64 \to 32$ | 2 | Leaky ReLU(0.2) |
| Concat3 (/w ResBlock1) | | $32 \to 64$ | | |
| Conv7 | $1 \times 1$ | $64 \to 32$ | 1 | Leaky ReLU(0.2) |
| Offset Block 3 | | $32 \to 32$ | | Leaky ReLU(0.2) |
| RSAB 3 | | $32 \to 32$ | | Leaky ReLU(0.2) |
| Concat4 (/w input) | | $32 \to 35$ | | |
| ConvOut | $3 \times 3$ | $35 \to 3$ | 1 | Linear |

## 3.2 Datasets

The original paper uses SIDD Medium [9] and RENOIR [10] datasets during training for denoising real noisy images and reports the qualitative and quantitative results on DND [11] and SIDD validation [9] datasets. Since the author's code only provides the data loader for synthetic image datasets, we have integrated our SIDD validation data loader implementation and the DND test script provided by TU Darmstadt [11] to our pipeline. For synthetic noise removal, additive white Gaussian noise with standard deviation of 30, 50 and 70 have been added to DIV2K dataset [12], which consists 800 high resolution images. To validate the performance of SADNet on synthetic noise removal task, the models are tested on BSD68 [13] and Kodak24 [14] datasets processed with the same noise addition mechanism.

Both test and validation data for all settings are composed of high resolution images. Therefore, they are fed to the model as 128x128 patches cropped by fixed coordinates, as described in the paper. We have applied $90°$ rotation, horizontal and vertical flipping to the images during training, following the practice in the paper.

## 3.3 Hyperparameters

In our replication study, we used the ADAM optimizer [16] with $\beta_1 = 0.9$, $\beta_2 = 0.999$, and $\epsilon = 1e - 8$, with an initial learning rate of $1e - 4$ during training, as described in the paper. The provided code initializes the weights of the convolutional layers in all blocks with Xavier Uniform method [23]. Since this choice has not been discussed in the paper, we left each convolution layer initialized by the default weight initialization method in PyTorch (*i.e.* Kaiming [24]) in our experiments.

## 3.4 Experimental setup

In this study, we have followed the same training procedures for all setting, and employed SSIM and PSNR values as performance metrics, as described in the original paper. The parameters for all training settings can be found in the

configuration file in our GitHub repository. Our implementation and the trained weights are open-sourced, and can be accessed at `https://anonymous.4open.science/r/96f7ac9a-be7b-4d7a-ac10-44da93e6948c/`.

## 3.5 Computational requirements

The experiments have been conducted on a single RTX 2080Ti for approximately 3 days, and only requires high GPU memory, mostly due to modulated deformable convolutions. It requires ~3GB GPU memory for training, and ~8GB CPU memory for loading the data, due to the file structure of datasets.

Table 2: Quantitative results on SIDD sRGB validation dataset. Compared methods: CBM3D [18], CDnCNN-B [19], CBDNet [4], PD [3], RIDNet [5], SADNet [1].

| Method | CBM3D | CDnCNN-B | CBDNet | PD | RIDNet | SADNet | SADNet (ours) |
|---|---|---|---|---|---|---|---|
| Blind/NonBlind | Non-blind | Blind | Blind | Blind | Blind | Blind | Blind |
| PSNR | 30.88 | 26.21 | 30.78 | 32.94 | 38.71 | **39.46** | 39.41 |

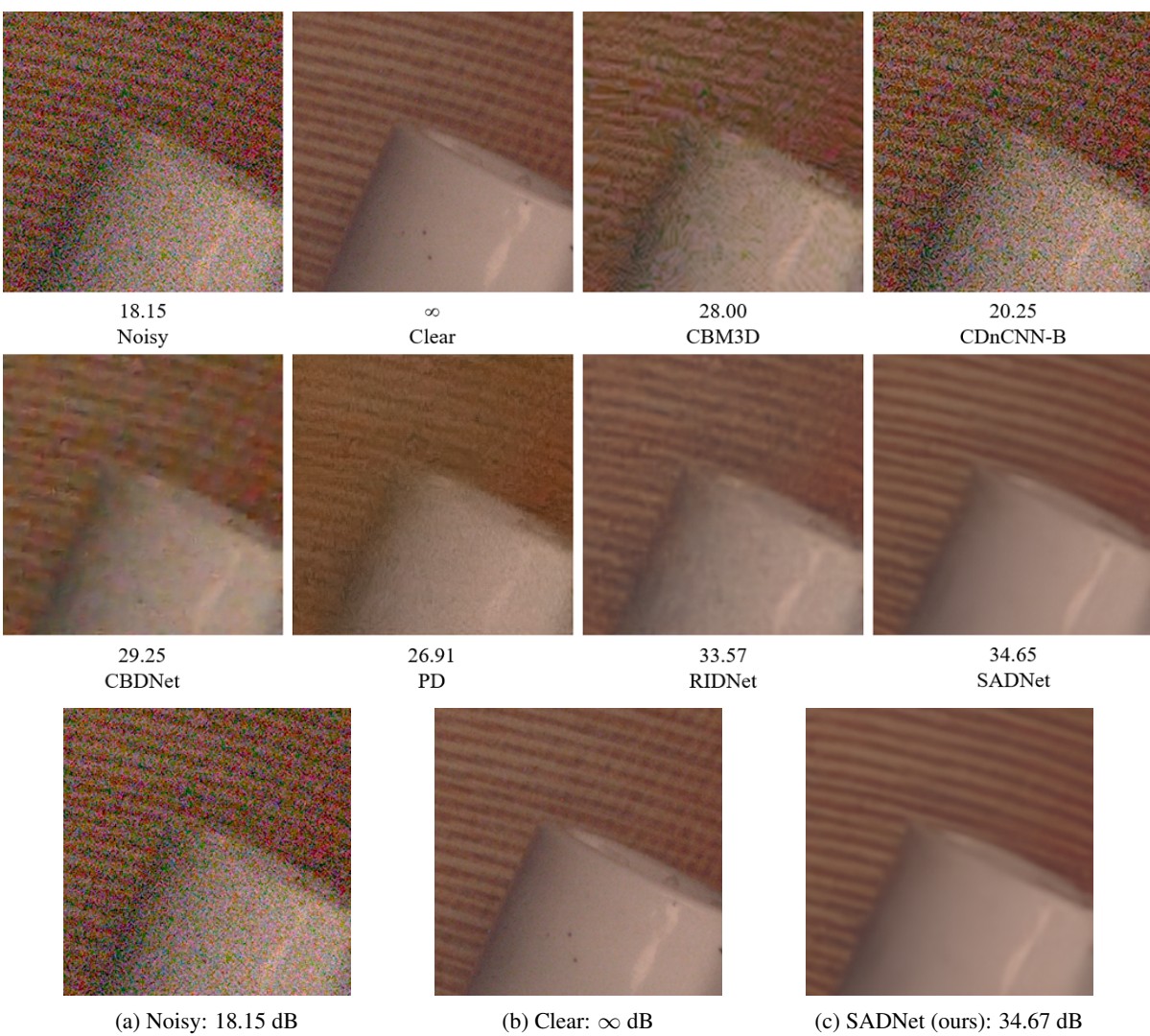

Figure 2: Real image denoising results on SIDD validation dataset. The results on the first two rows are obtained from the paper [1], the third row represents our results on the same image.

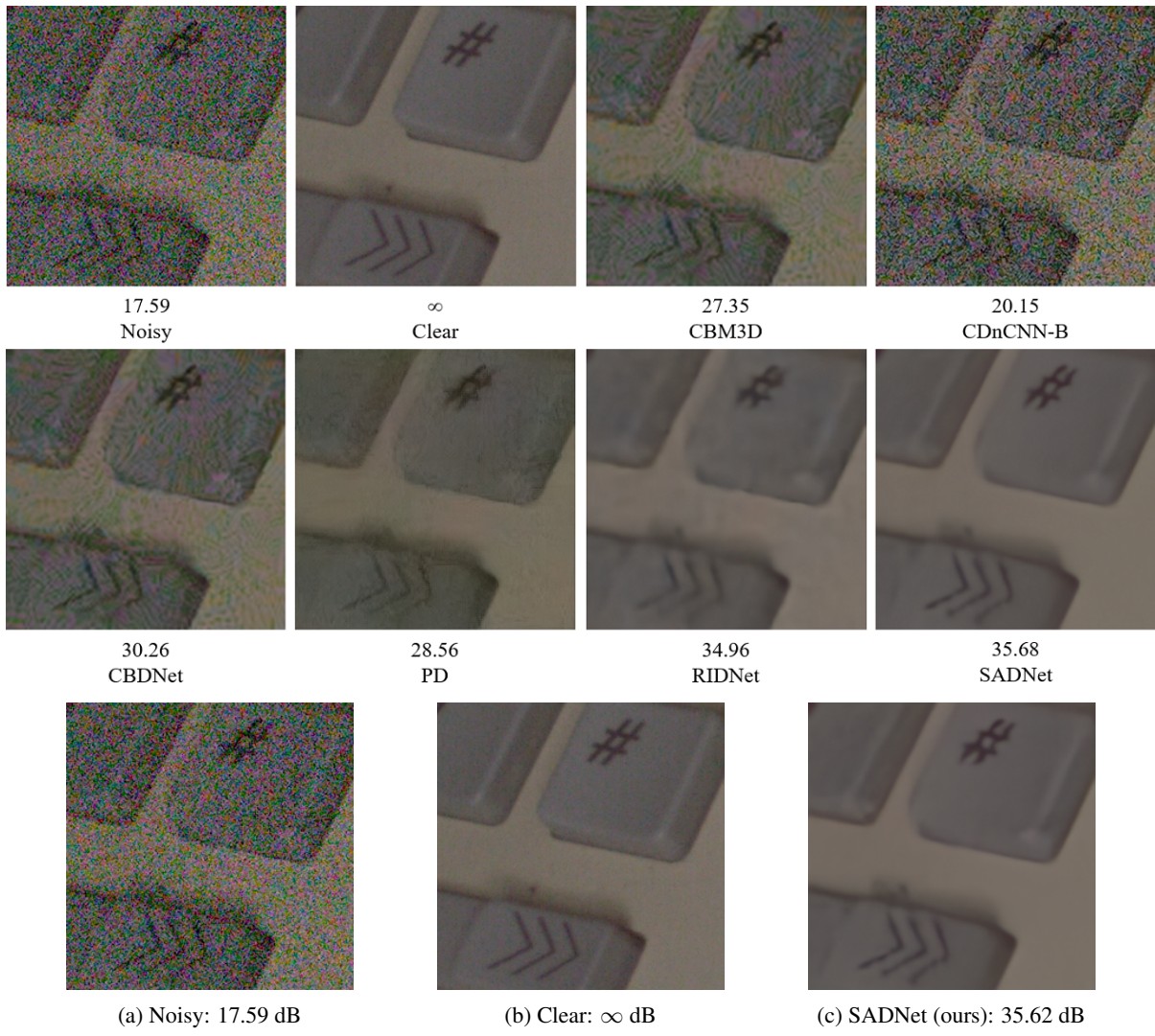

| 17.59 | ∞ | 27.35 | 20.15 |
| Noisy | Clear | CBM3D | CDnCNN-B |

| 30.26 | 28.56 | 34.96 | 35.68 |
| CBDNet | PD | RIDNet | SADNet |

| (a) Noisy: 17.59 dB | (b) Clear: ∞ dB | (c) SADNet (ours): 35.62 dB |

Figure 3: Real image denoising results on SIDD validation dataset. The results on the first two rows are obtained from the paper [1], the third row represents our results on the same image.

## 4 Results

We have implemented the model from scratch by following the descriptions presented in the original paper, and then achieved to replicate the claimed results by referring to the published code. Overall, our implementation of SADNet achieved on-par performances in SSIM and PSNR metrics on test datasets, and we also validated the results on both denoising tasks by examining their qualitative results.

As shown in Table 2, our quantitative results on SIDD sRGB validation dataset has 39.41 PSNR value, which is only 0.12% less than the one reported in the original paper. Moreover, the average duration of a single inference of SADNet is 26.7 ms. according to the paper, while our implementation of SADNet completes the single inference on 25.9 ms. The visual comparisons of real noise removal of the images are shown in Figure 2 and Figure 3. The samples are from SIDD validation dataset, according to the ones reported in the paper. The first two rows in these figures are directly taken from the original paper for further comparison with our replication results, which can be seen in the third row. On the results from the original paper, SADNet mainly differs from the compared methods by generating a distinct clear continuous stripe texture on the background while preserving the object surface appearance. Our replication clearly shares the similar behaviour. Therefore, we can state that the replicated quantitative results on real noise removal task are competitive enough, and also supports the main claim in the original paper.

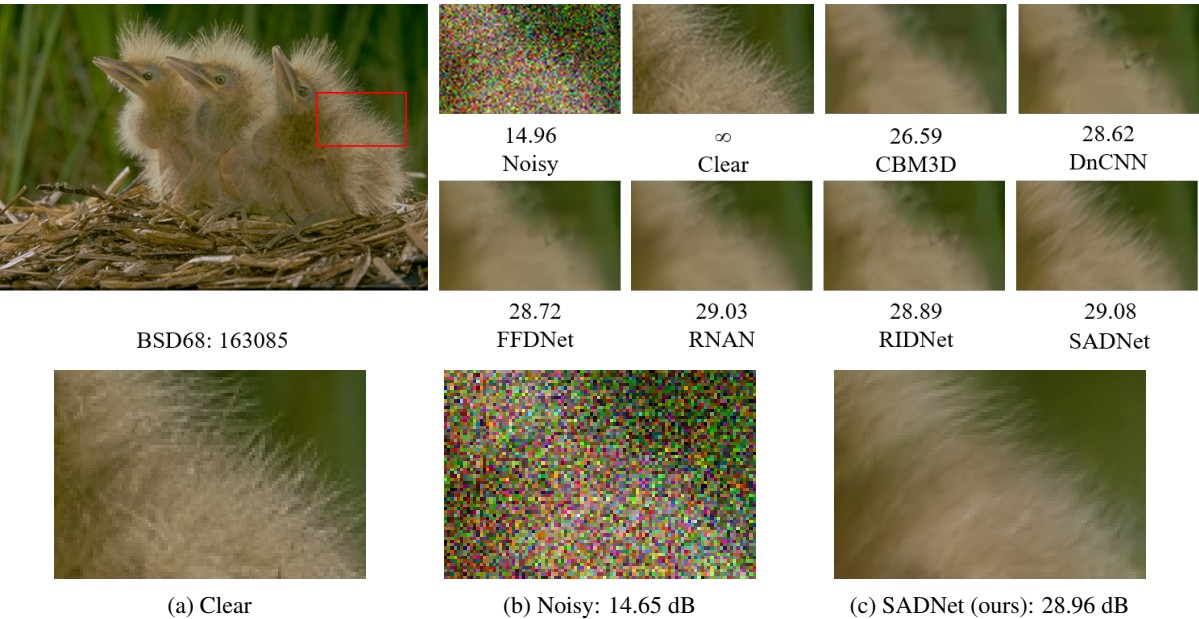

Figure 4: Synthetic image denoising results on BSD68 dataset with noise level $\sigma = 50$. The results on the first row are obtained from the paper [1], the second row represents our results on a similar patch.

Similarly, Table 3 demonstrates that the results of our SADNet implementation achieves on-par PSNR values with the ones reported in the paper for different noise levels (*i.e.* $\sigma \in \{30, 50, 70\}$) on BSD68 and Kodak24 datasets. Particularly, we have obtained better results than all other compared methods and the reported SADNet results on Kodak24 dataset for all noise levels. Moreover, the replicated SADNet model imitates the qualitative results of the original model on both datasets. Although the images from Kodak24 and BSD68 are heavily exposed to the synthetic noise, SADNet has the ability to remove noise, and to generate well-defined textures when compared to the recent works. As shown in Figure 4, all other compared methods have smoothed the texture and swept away the feather details, meanwhile the original implementation of SADNet and ours achieve to generate more plausible feather-like texture. Similar to the previous example, in Figure 5, the clothing details are significantly preserved, especially pilling on the top-left part of the cloth and the vertical texture details on the cloth.

The ground truth of DND validation set is private, and thus it is not possible to locally validate the results on this dataset. Despite of several attempts to submit our results to DND online validation system, we could not obtain SSIM and PSNR results, due to the server error. We have tried to contact with DND Team, but we could not get any advice for solving this issue.

Table 3: Average PSNR(dB) results on synthetic color noisy images.

| Datasets | Kodak24 ($\sigma$) | | | BSD68 ($\sigma$) | | |
|---|---|---|---|---|---|---|
| | 30 | 50 | 70 | 30 | 50 | 70 |
| **Models** | | | | | | |
| **CBM3D** [18] | 30.89 | 28.63 | 27.27 | 29.73 | 27.38 | 26.00 |
| **DnCNN** [19] | 31.39 | 29.16 | 27.64 | 30.40 | 28.01 | 26.56 |
| **MemNet** [20] | 29.67 | 27.65 | 26.40 | 28.39 | 26.33 | 25.08 |
| **FFDNet** [21] | 31.39 | 29.10 | 27.68 | 30.31 | 27.96 | 26.53 |
| **RNAN** [22] | 31.86 | 29.58 | 28.16 | 30.63 | 28.27 | 26.83 |
| **RIDNet** [5] | 31.64 | 29.25 | 27.94 | 30.47 | 28.12 | 26.69 |
| **SADNet** [1] | 31.86 | 29.64 | 28.28 | **30.64** | **28.32** | **26.93** |
| **SADNet (ours)** | **32.06** | **29.86** | **28.47** | 30.58 | **28.30** | 26.91 |

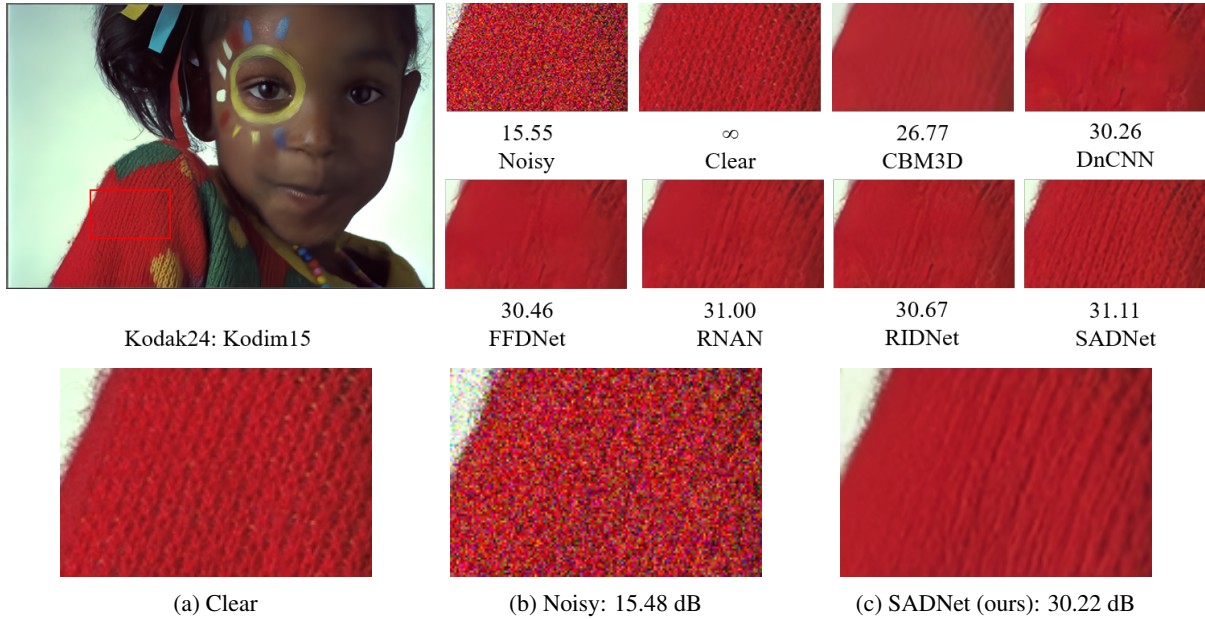

Figure 5: Synthetic image denoising results on Kodak24 dataset with noise level $\sigma = 50$. The results on the first row are obtained from the paper [1], the second row represents our results on a similar patch.

## 5 Discussion

The qualitative results generated with our replication strongly resemble to the presented results, and differs from the other compared studies. According to these results, we can state that our implementation of SADNet consistently yields visually-plausible results on both real and synthetic noisy images, and supports the claims of the original paper. In addition, our experiments firmly correlates with the reported PSNR values.

Overall, the paper and the provided code was sufficient for replicating the results on real and synthetic noise removal. For re-implementing the model from scratch, we have only referred to the paper, and ended up on-par performance with the ones in the paper on all settings.

Lastly, to provide an insight for run-time on different hardware, our replication has 25.9 ms. inference run-time on real noise removal task, whereas the reported run-time duration is 26.7 ms. Note that we used a single RTX 2080Ti GPU during our experiments, while a single GTX 1080Ti GPU is used in the original study, and we assume that this is the reason of this difference.

### 5.1 What was easy

The code was open-source, and implemented in PyTorch, hence adopting the training loop and proposed blocks to our implementation facilitated our reproduction study. The loss function is straightforward and the architecture has a *U-Net-like* structure, so that we could achieve to implement the architecture in a fair time.

### 5.2 What was difficult

Due to the lack of compatibility with the current versions of PyTorch and TorchVision and the dependency on an external CUDA implementation of deformable convolutions, we have encountered several issues during our implementation. Then, we have considered to re-implement residual spatial-adaptive block and context block from scratch for deferring these dependencies, however, we could not achieve this just by referring to the paper. Therefore, we have decided to directly use the provided blocks as in the author's code.

### 5.3 Communication with original authors

We did not make any contact with the authors since we achieved to solve the issues encountered during the implementation of SADNet by examining the author's code.

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
