# OpenReview forum: "[Re] Spatial-Adaptive Network for Single Image Denoising"
_ML_Reproducibility_Challenge/2020 — RC2020_

### Official Review · AnonReviewer1 · 2021-02-25
**A reproducibility study of the SADNet denoising model**

**Rating:** 8
**Confidence:** 4

**Review:**

The given report examines the reproducibilty of a deep denoising model SADNet for natural images based on spatial-adaptive residual blocks. The model was proposed in the paper "Spatial-Adaptive Network for Single Image Denoising" by Chang et al. The model was reimplemented by the authors of the report using the residual spatial-adaptive block and the context block from Chang et al.

**Reproducibility Summary:**
The report contains a reproducibility summary as required by the template.

**Scope of Reproducibility:**
The authors of the report clearly state the scope of their reproducibility experiments.

**Code:**
The code was provided. The authors reimplemented the training loop and the model and reused some code from the original repository (the residual spatial-adaptive block and the context block). The authors use the dataloader provided with the dataset. Overall, the code is well written but some comments and doc-strings would highly improve the readability. Unfortunately, I was not able to run the code due to CUDA incompatibilities.

**Communication with Original Authors:**
The authors of the report state that no communication with Chang et al. was necessary to reproduce the results.

**Hyperparameter Search:**
The authors use the hyperparameter settings stated in the paper. No further hyperparameter sweeps were performed. Given the long training time of three days this choice seems reasonable. However, the authors changed the initialization scheme to the Kaiming initializer (instead of Xavier uniform initializer).

**Ablation Study:**
No ablation study was carried out.

**Discussion on Results:**
The authors were able to reproduce most of the results from the original paper. The PSNR/SSIM values closely match the values reported by Chang et al. However, visually the results shown in Figure 5 are qualitatively worse than the original results. Also runtimes are compared with the runtime stated in the original work (using different hardware) which is not appropriate. Running the original code on the same hardware would have been a more meaningful way of comparing runtimes.

**Recommendations for Reproducibility:**
No recommendations made.

**Results Beyond the Paper:**
No results beyond the original work reported.

**Overall Organization and Clarity:**
The report is well-written and well-organized. The authors should add references for the first sentence of the introduction. Some small comments on the writing can be found at the end of the review.

**Summary of Review:**
The report at hand successfully reproduced the original paper. The authors are very clear about their experimental setup and the problems that they faced during reproduction. However, there are some minor problems in the discussion. Nonetheless, I recommend to accept the paper to the ML Reproducibility Challenge 2020.

**Minor Remarks:**

- line 90: recieves -> receive
- line 114-115: 1e-4 vs 10^8
- caption Figure 2 and Figure 3: validation dataset should not be capitalized
- Equation (1): the \delta in front of m_i is missing
- Equation (2): set-notation is not appropriate here: \delta p^s and \delta m^s should form a tuple instead of a set. However, this notation is also used in the original work.
- line 144: \sigma = {30, 50, 70}: here \in should be used instead of "="


**Familiar With The Original Paper:**

I have read the original paper

**Reproducibility Summary:**

Report has summary

---

### Official Review · AnonReviewer2 · 2021-02-28
**SADNet reimplementation**

**Rating:** 7
**Confidence:** 3

**Review:**


> Overall approach to reproduce the results and the adjoining paper are quite clear.

> The reproducibility summary is provided.

> It seemed easy to reproduce the results from the paper although the authors had to resort to default hyperparameters as these were not readily available. Their results still tallied with that of the original paper showing that the model architecture is robust to change in hyperparameter settings.

> It is not clear if the authors rewrote the original PyTorch code.

> Even though no communication with the original authors was made, some areas to tidy up in the code are discussed by the authors.



**Familiar With The Original Paper:**

I have not read the original paper

**Reproducibility Summary:**

Report has summary

---

### Decision · Program_Chairs · 2021-03-31

**Decision:**

Accept

**Comment:**

Selected for ReScience-C Journal Publication.

Good reviews, building the code from scratch is an important contribution.